# Partially randomised patient preference trials as an alternative design to randomised controlled trials: systematic review and meta-analyses

Karin A Wasmann  ,[1] Pieta Wijsman,[2] Susan van Dieren,[3] Willem Bemelman,[1] Christianne Buskens[1]

[1]Department of Surgery, Amsterdam UMC–Location AMC, Amsterdam, Netherlands
[2]Department of Internal Medicine, Spaarne Gasthuis, Haarlem, Netherlands
[3]Department of Statistics and Epidemiology, Amsterdam UMC–Location AMC, Amsterdam, Netherlands

**Correspondence to**
Dr Karin A Wasmann;
k.a.wasmann@amsterdamumc.nl

## ABSTRACT

**Objective** Randomised controlled trials (RCT) are the gold standard to provide unbiased data. However, when patients have a treatment preference, randomisation may influence participation and outcomes (eg, external and internal validity). The aim of this study was to assess the influence of patients' preference in RCTs by analysing partially randomised patient preference trials (RPPT); an RCT and preference cohort combined.

**Design** Systematic review and meta-analyses.

**Data sources** MEDLINE, Embase, PsycINFO and the Cochrane Library.

**Eligibility criteria for selecting studies** RPPTs published between January 2005 and October 2018 reporting on allocation of patients to randomised and preference cohorts were included.

**Data extraction and synthesis** Two independent reviewers extracted data. The main outcomes were the difference in external validity (participation and baseline characteristics) and internal validity (lost to follow-up, crossover and the primary outcome) between the randomised and the preference cohort within each RPPT, compared in a meta-regression using a Wald test. Risk of bias was not assessed, as no quality assessment for RPPTs has yet been developed.

**Results** In total, 117 of 3734 identified articles met screening criteria and 44 were eligible (24 873 patients). The participation rate in RPPTs was >95% in 14 trials (range: 48%–100%) and the randomisation refusal rate was >50% in 26 trials (range: 19%–99%). Higher education, female, older age, race and prior experience with one treatment arm were characteristics of patients declining randomisation. The lost to follow-up and crossover rate were significantly higher in the randomised cohort compared with the preference cohort. Following the meta-analysis, the reported primary outcomes were comparable between both cohorts of the RPPTs, mean difference 0.093 (95% CI −0.178 to 0.364, p=0.502).

**Conclusions** Patients' preference led to a substantial proportion of a specific patient group refusing randomisation, while it did not influence the primary outcome within an RPPT. Therefore, RPPTs could increase external validity without compromising the internal validity compared with RCTs.

**PROSPERO registration number** CRD42019094438.

### Strengths and limitations of this study

► This systematic review and meta-analyses of partially randomised patient preference trials (RPPT) provides unique data on external and internal validity between randomised and patients' preference cohorts.
► It was not possible to objectively establish the quality of included trials, as there is currently no valid critical appraisal tool to apply for an RPPT.
► Uniform counselling is of crucial importance in RPPTs, which has not been standardly reported in the included studies.

## INTRODUCTION

Randomised controlled trials (RCT) are suggested to provide the most reliable evidence for treatment efficacy.[1] However, participants are no passive recipients of interventions. Patients with a treatment preference may decline enrolment to avoid being randomised to their non-preferred treatment. Consequently, treatment preferences can decrease the generalisability of RCT results to the clinical population (ie, reduce external validity). Additionally, trials comparing experimental versus standard treatment are likely to include patients preferring experimental treatment, as trial participation is not needed for patients preferring standard treatment, further reducing external validity. Internal validity may be reduced, as randomisation to the (non-) preferred strategy could influence adherence to treatment protocol and study outcomes. Subjective study outcomes can directly be affected by treatment preference, whereas objective outcomes are most likely affected indirectly via adherence (so-called reluctant acquiescence phenomenon). Especially for an unblinded trial comparing treatments of significant different nature (eg, medical vs surgical) the RCT could be

BMJ

an inappropriate design. Throughout the years, several approaches using various names have been proposed as alternative designs to diminish the influence of patients' preference on validity: a partially randomised patient preference trial (RPPT), a comprehensive cohort trial, a patient preference trial, and more.[2] In general, the aim of these designs is to treat patients with a preference for treatment strategies accordingly, whereas only those patients without a distinct preference will be randomised in the usual way.[3] In the era of patients becoming more active participants in research, the use of RPPTs increases. The two previous systematic reviews addressing influence of preference on validity concluded that this influence was limited.[4 5] However, one review only included studies addressing psychotherapy, and the other dates from 2005. So far, the value of the RPPT remains unclear, nor has it been addressed in the Oxford Levels of Evidence (Centre for Evidence-Based Medicine).[6]

The aim of the study was to assess the influence of patients' preference following randomisation in current daily clinical practice, by comparing randomised cohorts with preference cohorts within all RPPTs published since 2005. Two hypotheses were tested: (1) Patients' preference will negatively influence participation in RCTs, decreasing external validity. Therefore, the external validity of an RPPT will be higher. (2) Patients' preferences will influence adherence and outcomes in RCTs, decreasing internal validity. However, as only the remaining indifferent patients will be included in the RCT cohort of an RPPT, this RCT cohort can be considered as the true gold standard for internal validity.

## METHODS

### Design

A systematic review and meta-analyses of RPPTs was conducted. This study is reported in accordance with the Cochrane Handbook for Systematic Reviews of Interventions[7] and the Preferred Reporting Items for Systematic Reviews and Meta-Analyses (PRISMA) statement (online supplementary material 1).[8] The study protocol is available in online supplementary material 2.

### Data sources and searches

A search in PubMed, Embase, PsycINFO and the Cochrane Library for RPPTs published between 1 January 2005 and 5 October 2018 was executed without language restriction with the assistance of a librarian. The subject in the search strategy was RPPT and possible aliases of RPPT (see the PubMed Search Strategy section). Database searches were supplemented by hand searching reference lists of relevant articles. Additionally, authors were contacted to seek for data from unpublished studies identified. Non-English language articles were translated for possible inclusion.

### PubMed search strategy

5 October 2018

(patient preference design*[tiab] OR patient preference model*[tiab] OR patient preference trial*[tiab] OR patient preference method*[tiab] OR comprehensive cohort stud*[tiab] OR comprehensive cohort design*[tiab] OR patient preference group[tiab] OR patient preference allocation arms[tiab] OR preference allocation[tiab] OR randomized preference trial*[tiab] OR randomised preference trial*[tiab] OR preference arms[tiab] OR preferences[ti] OR treatment preference basis[tiab] OR (patient preference*[tiab] AND random*[ti]) OR (prefer*[ti] AND random*[ti]) OR (registry patient*[tiab] AND randomized[tiab])) AND ("Clinical Trial"[pt] OR trial[ti] OR preference trial[tiab]) AND ("2004/09"[Date - Publication] : "3000"[Date - Publication])

And

((patient preferences[ti] AND clinical trials[ti]) OR nonrandomized[ti] OR (patient preference[ti] AND randomization[ti]) OR (random[ti] AND nonrandom assignment[ti]) OR (randomized[ti] AND non-randomized[ti]) OR (nonrandom assignment[ti]) OR (randomized[ti] AND nonrandomized[ti]) OR (randomi*[tiab] AND preference arm) OR (partially randomized study[tiab] AND "Randomized Controlled Trial"[pt]) OR (unwilling to be randomized[tiab] AND "Randomized Controlled Trial"[pt]) OR (choice[tiab] AND randomisation[tiab] AND "Randomized Controlled Trial"[pt])) AND (random*[tiab]) AND ("Clinical Trial"[pt] OR trial[ti] OR clinical trials[ti]) AND ("2004/09"[Date - Publication] : "3000"[Date - Publication])

"comprehensive cohort*"[tiab] AND ("2004/09"[Date - Publication] : "3000"[Date - Publication])

### Study selection

RPPTs describing results of both the randomised and preference cohorts, as long as in both cohorts patients met the same inclusion and exclusion criteria and were treated according to the same treatment protocol, were included. Trials in which a two-stage randomised design was conducted, allocation was based on doctors' preference, without available separate data for the randomised and preference cohorts, with economic primary outcomes, or with non-clinical populations were excluded. Furthermore, it was decided not to include older RPPTs (before 2005), as it is important to consider the value of this design for current daily practice. A previous systematic review addressing on the value of RPPTs was published in 2005, which can be used to interpret results from older studies.[4]

### Data extraction

The two first authors independently screened the citations and abstracts for eligible articles using a prepiloted standardised data form (Covidence; Veritas Health Innovation, Melbourne, VIC, Australia). Disagreements were discussed at steering group meetings.

The same two authors extracted data with the use of the same data form. Multiple publications reporting on the same trial were considered as one single trial for these analyses.

The level of sought data was summary estimates. Authors were contacted for further information when necessary. In case they were not forthcoming, the study was included in the review, but excluded from our reanalysis and/or meta-analyses.

### Risk of bias assessment

Quality assessment of the trials was not performed, as no quality assessment for RPPTs has yet been developed and current criteria predominantly relate to concealment of randomisation (eg, Risk of Bias in Non-Randomized Studies-I and Cochrane Risk of Bias); consequently quality assessment and variability between trials were not applicable.[9][10] Since the outcomes of each trial greatly differed, also the risk of bias assessment for systematic reviews (eg, Grading of Recommendations, Assessment, Development and Evaluations) was not applicable.[11]

### Outcomes

The primary outcomes were external and internal validity between randomised and preference cohorts within RPPTs. To analyse whether patients' preference influenced external validity, data were extracted on participation rates in the randomised and preference cohorts. To assess if a specific patient group accepted randomisation, data were extracted on baseline characteristics of the randomised and preference cohorts of an RPPT separately. These characteristics were categorised into sociodemographic and clinical factors. Subsequently, these factors were compared between the randomised and preference cohorts of RPPTs.

To analyse whether patients' preference influenced internal validity, data were extracted on lost to follow-up, crossovers and primary outcomes of the randomised and preference cohorts of an RPPT separately. Subsequently, these outcomes were compared between the randomised and preference cohorts within RPPTs. The primary outcomes of RPPTs were identified through explicit statements, study hypotheses, reported power analyses, and were checked on similarity with the study protocol. If this was not sufficient, the most likely primary outcome was chosen by consensus (KAW and SvD), or the study was excluded. To compare the primary outcomes between the randomised and preference cohorts within RPPTs, the outcome effects were compared between the randomised cohort and the preference cohort. It is emphasised that comparisons of outcome between randomised and preference cohorts are subject to bias, and if not done by the study itself, it was not possible to adjust for confounding factors. If in studies the adjusted and non-adjusted primary outcomes were available, the adjusted outcomes were used. Subsequently, separate analyses on adjusted and non-adjusted primary outcomes were performed.

### Statistical analysis

The randomisation rate, participation rate and difference in baseline characteristics between the randomised and preference cohorts were explored and described, but not compared using statistics. To assess differences in baseline characteristics, mean and SDs were compared. If median IQRs were reported, it was converted to mean and SDs.[12] When baseline characteristics were presented per experimental and control group, the sum of mean and SDs of these two groups was calculated for the randomised and preference cohorts using a weighted t-test. The lost to follow-up and cross-over rates were compared using a random effects model meta-analysis for proportions.

To realise the comparison of the primary outcomes of randomised and preference cohorts, a reanalysis was conducted. Because the trials involved a range of diseases, outcome measures and sample sizes, different treatment effect scales were converted into standardised effect sizes in the reanalysis. Treatment effects were calculated directly for continuous outcome variables as standardised mean differences (difference in means divided by the pooled SD). For binary outcomes, log ORs were calculated and converted into standardised effect size differences.[13] In case none of the patients in the preference cohort chose the control treatment, the treatment effect of the experimental treatment was compared with the control treatment of the randomised cohort. Only trials for which a 'net' effect (primary outcome minus baseline value of the primary outcome) could be calculated were included in the meta-analyses. In case the 'net' effect was missing, but baseline values and primary outcomes were available, the SD was estimated.[14] Heterogeneity was not assessed as trial outcomes were different for each study included. Meta-analysis of randomised versus preference cohort was performed using a random effects model with an inverse variance weighting. A final meta-regression was performed using a Wald test to compare the standardised treatment effects.

A $p < 0.05$ was considered a significant difference. R's programming environment was used (V.3.5.1, R Foundation for Statistical Computing, Vienna, Austria).

### Patient and public involvement

There was no direct involvement of patients or the public in the development of the research question, selection of the outcome measures, design and implementation of the study, or interpretation of the results.

### RESULTS

In total, 117 out of 3734 records identified were full text screened. Fifty-eight partially RPPTs from 2005 onwards were found, of which 44 (including 24 873 patients) were eligible for at least basic data extraction (table 1), and 20 could be included in the meta-analyses (PRISMA flow chart, figure 1).[15–72] Exclusion reasons for the meta-analyses were: no availability of both treatment outcomes in the randomised and preference cohorts separately in 14 trials,[15 16 18 19 23 24 27 30 31 34 39 41 42 63] no availability of SDs, which could also not be converted from other

**Table 1** Partially randomised patient preference trials included in the review

| Source | Population | R (n) | P (n) | Field | Intervention and comparison groups | Primary outcome(s) |
|---|---|---|---|---|---|---|
| Ashok et al[15] | Women presenting for termination of pregnancy | 400 | 86 | Gynaecology | Medical versus surgical termination*† | Acceptability at 2 weeks |
| Barnard et al[16] | Premenopausal women with symptomatic uterine fibroids | 59 | 34 | Gynaecology | UAE versus MRgFUS*† | Perioperative outcomes at 3 months |
| Bergk et al[18] | Patients with DSM-IV disorder | 27 | 81 | Psychiatry | Mechanical restraint versus seclusion | CES at 4 weeks |
| Boers et al[19] | Pregnant women with disproportional intrauterine growth | 650 | 452 | Gynaecology | Induction versus expectative monitoring* | (S)AE neonate at discharge |
| Brinkhaus et al[20]‡ | Patients with allergic asthma | 357 | 1088 | Social medicine | Acupuncture versus control* | AQLQ at 3 months |
| Brinkhaus et al[21] | Patients with allergic rhinitis | 981 | 4256 | Social medicine | Acupuncture versus control* | RQLQ at 30 days |
| Buhagiar et al[22]‡ | Patients after total knee arthroplasty | 165 | 87 | Orthopaedics | Home-based versus inpatient recovery | Walking distance at 36 weeks |
| Chekerov et al[23] | Elderly with ovarian cancer receiving chemotherapy | 3 | 116 | Gynaecology | Oral versus intravenous treosulfan | DFS at 2 years |
| Creutzig et al[24] | Paediatric patients with relapsed AML | 101 | 54 | Haematology | L-DNR/FLAG versus FLAG | OS at 4 years |
| Crowther et al[25] | Pregnant women with one prior caesarean | 22 | 2323 | Gynaecology | Caesarean versus vaginal birth*† | Death and SAE at 30 days |
| Dalal et al[26]‡ | Participants in cardiac rehabilitation after acute MI | 104 | 126 | Cardiology | Home-based versus hospital recovery | HADS at 9 months |
| Ejlertsen et al[27] | Premenopausal patients with breast cancer | 525 | 1628 | Oncology | Chemotherapy versus ovarian ablation*† | DFS at 10 years |
| Erkan et al[28] | Patients with positive aPL but no vascular and/or pregnancy events. | 98 | 74 | Internal medicine | Aspirin versus placebo or no aspirin* | Acute thrombosis per 100 patient-years |
| Fong et al[29] | Patients with adolescent idiopathic scoliosis | 19 | 50 | Orthopaedics | Brace versus observational* | Recruitment feasibility |
| Gall et al[30] | Patients undergoing colon cancer surgery | 203 | 135 | Surgery | GP versus surgeon follow-up | PCS score at 24 months |
| Glazener et al[31] | Patients with vaginal wall prolapse | 1348 | 1126 | Gynaecology | Mesh versus no mesh*† | POPSS at 12 months |
| Grant et al[32]‡ | Patients with gastro-oesophageal reflux disease | 357 | 453 | Upper GI | Surgery versus medication*† | Reflux QoL at 1 year |
| Hatcher et al[34] | Patients presenting with self-harm | 552 | 542 | Psychiatry | PST plus standard care versus standard care* | Repeated self-harm at 1 year |
| Howard et al[35]‡ | Women requiring voluntary psychiatric admission | 42 | 61 | Psychiatry | Crisis houses versus psychiatric wards | Functioning (GAF) at 12 weeks |
| Hubacher et al[36]‡ | Women aged 18–29 years who were seeking a short-acting method | 382 | 524 | Gynaecology | Long-acting versus short-acting contraceptive* | Continuation rate at 1 year |
| Jones et al[37]‡ | Patients with palliative cancer | 41 | 36 | Oncology | Advance versus usual care* | VAS(S) at 8weeks |
| Karlsen et al[39] | Patients with proximal ureter stones | 50 | 21 | Urology | Shock wave versus ureteroscopy*† | Stone-free rate at 3 months |
| Kearney et al[40] | Patients with an acute Achilles tendon rupture | 20 | 29 | Orthopaedics | Surgery versus conservative*† | Disability rating index at 9 months |
| Kröz et al[41] | Patients with breast cancer-related fatigue | 65 | 61 | Oncology | Multimodel combined programme versus aerobic training* | PSQI at 10weeks |
| Lock et al[42] | Children with recurrent sore throats | 268 | 461 | Children surgery | Surgery versus medication*† | Number of episodes of sore throat at 2 years |
| Majumdar et al[43]‡ | Patients with lower urinary tract symptoms (LUTS) | 99 | 210 | Urology | Urodynamics versus conservative*† | King's QoL at 6 months |
| Mittal et al[46]‡ | Patients with type B ankle fracture | 160 | 276 | Orthopaedics | Surgery versus no surgery*† | FAOQ and PCI at 12 months |
| Prescott et al[49] | Women after breast-conserving surgery | 255 | 100 | Oncology | Non-radiotherapy versus radiotherapy* | QoL after 5 years |
| Purepong et al[50]‡ | Office workers suffering from low back pain (LBP) | 64 | 37 | Physical therapy | Backrest versus no intervention* | VAS at 3 months |
| Raue et al[52] | Patients operated for diverticulitis | 149 | 294 | Surgery | Laparoscopic versus open approach | QoL at 30 days |
| Robson et al[53]‡ | Termination of pregnancy less than 14 weeks' gestation | 349 | 1528 | Gynaecology | Medicine versus surgery TOP*† | Acceptability TOP at 2 weeks |
| Schweikert et al[55] | Patients for cardiac rehabilitation | 4 | 163 | Cardiology | Outpatient versus inpatient recovery | EQ-5D at 12 months |
| Shi et al[58]‡ | Patients with vascular dementia | 48 | 20 | Complementary medicine | Acupuncture versus training* | SDSVD at 6 weeks |

Continued

**Table 1** Continued

| Source | Population | R (n) | P (n) | Field | Intervention and comparison groups | Primary outcome(s) |
|---|---|---|---|---|---|---|
| Sinclair et al[59]‡ | Patients with severe lung disease | 67 | 82 | Pulmonology | Advance care planning versus standard | ACP uptake at 6 months |
| Schwieger et al[56]‡ | Adolescent with idiopathic scoliosis (AIS) | 132 | 187 | Orthopaedics | Brace versus observation* | QoL at 2 years |
| Underwood et al[60]‡ | Older patients with chronic knee pain | 282 | 303 | Orthopaedics | Topical versus oral ibuprofen | WOMAC at 12 months |
| van der Kooij et al[62] | Uterine fibroids | 177 | 103 | Gynaecology | Embolisation versus hysterectomy*† | HRQoL at 12 months |
| Van Heest et al[63] | Children with upper extremity cerebral palsy | 29 | 10 | Orthopaedics | Surgery versus botulinum therapy*† | SHUEE at 24 weeks |
| Weinstein et al[65]‡ | Patients with spondylolisthesis | 304 | 303 | Orthopaedics | Surgical versus non-surgical*† | Physical functioning (SF-36 Phys) at 2 years |
| Weinstein et al[64]‡ | Patients with spinal stenosis | 289 | 365 | Orthopaedics | Surgical versus non-surgical*† | Physical functioning (SF-36 Phys) at 2 years |
| Witbrodt et al[67]‡ | Addicted people | 293 | 321 | Social medicine | Community residential versus day hospital* | Abstinence at 12 months |
| Witt el al[68]‡ | Patients with chronic low back pain | 2841 | 8537 | Rheumatology | Acupuncture versus control* | HFAQ at 3 months |
| Witt et al[69]‡ | Patients with osteoarthritis | 712 | 2921 | Rheumatology | Acupuncture versus control* | Osteoarthritis index (WOMAC) at 3 months |
| Woodward and Kelly[72] | Pregnant women | 60 | 20 | Gynaecology | Water versus land birth | Baby condition at 6 weeks |

*These 32 trials compared interventions versus conservative treatment.
†These 16 trials compared surgical interventions versus conservative treatment.
‡These 20 trials could be used to calculate standardised effect sizes of the randomised and preference cohorts separately, and were included in our reanalysis on the effect of preference on outcome.
ACP, advance care planning;aPL, antiphospholipid; AQLQ, Asthma Quality of Life Questionnaire; CES, Coercion Experience Scale; DFS, disease-free survival;DSM-IV, Diagnostic and Statistical Manual of Mental Disorders, Fourth Edition; EQ-5D, EuroQol-5 Dimension; FAOQ, Foot and Ankle Outcomes Questionnaire; FLAG, fludarabine; GAF, Global Assessment of Functioning;GP, general practitioner; HADS, Hospital Anxiety Depression Scale; HFAQ, Hannover Functional Ability Questionnaire; HRQoL, health-related quality of life; L-DNR, liposomal daunorubicin;MI, myocardial infarction; MRgFUS, MRI-guided focused ultrasound surgery; OS, overall survival; P, preference; PCI, physical component score; PCS, peritoneal cancer score; POPSS, pelvic organ prolapse symptom score; PSQI, Pittsburgh Sleep Quality Index; PST, problem-solving therapy; R, randomised; RQLQ, Rhinitis Quality of Life Questionnaire; SAE, serious adverse event; SDSVD, scale of differentiation of syndromes of vascular dementia;SF-36, Short Form-36; SHUEE, Shriners Hospital Upper Extremity Evaluation; TOP, termination of pregnancy; UAE, uterine artery embolisation; VAS, visual analogue scale; WOMAC, Western Ontario and McMaster Universities Osteoarthritis Index.

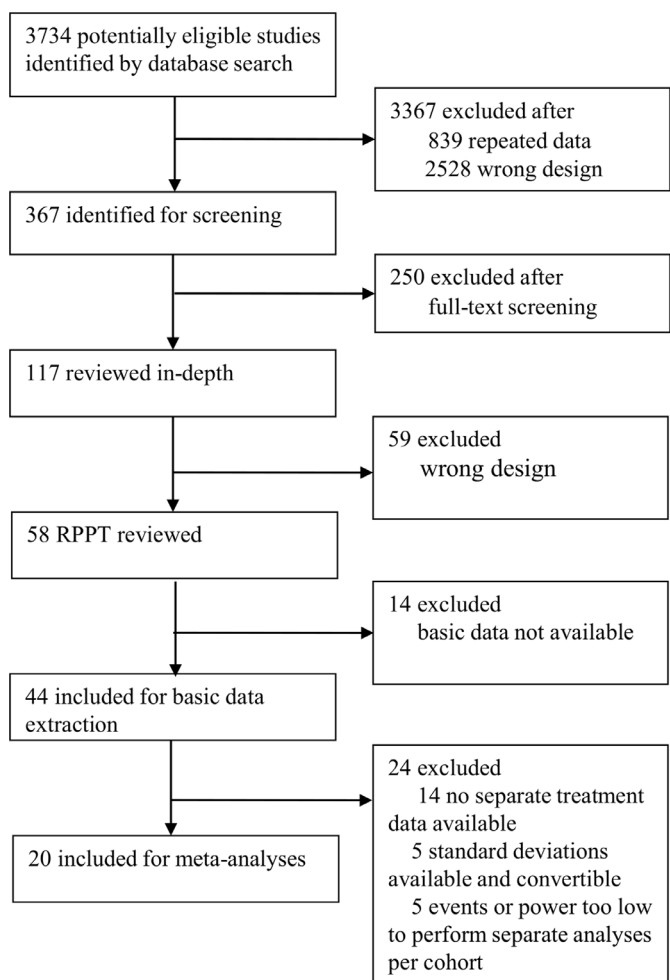

**Figure 1** Study selection according to Preferred Reporting Items for Systematic Reviews and Meta-Analyses (PRISMA). RPPT, randomised patient preference trial.

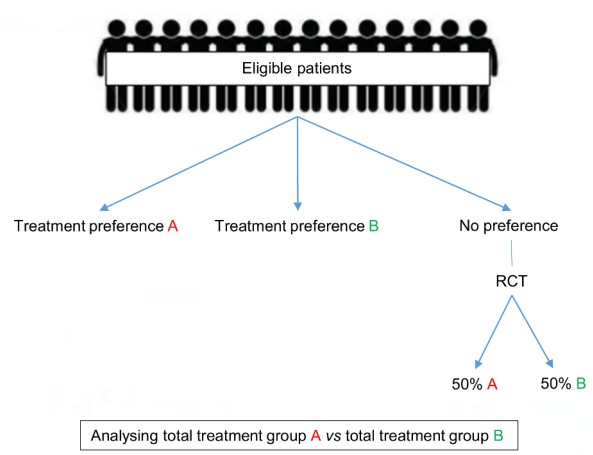

**Figure 2** A randomised patient preference trial. RCT, randomised controlled trial.

available data in five trials,[21 29 49 52 62] and the number of events or the power of one or both cohort(s) was too low to perform separate randomised and preference analyses in five trials.[25 28 40 55 72] The trials covered a wide range of clinical areas and interventions. The main areas were gynaecology (n=11), orthopaedics (n=10) and psychiatry (n=5). Of the 44 included trials, 32 compared an intervention versus conservative treatment, including 16 surgical interventions (table 1). In all trials but one, if patients refused randomisation they received their preference treatment (figure 2). In the other study, a Zelen randomisation was performed, randomising all eligible patients and afterwards asking for their consent to participate in the randomised arm or if they preferred the other intervention.[34] Parental preference was relevant in five trials involving children, as permission of parents was required and the preference between patients and parents could not be distinguished.[24 29 42 56 63]

**External validity**
The following results concern the influence of patients' preference on external validity. Information on the

number of eligible patients who agreed to participate (in either the randomised or preference cohort) was available in 39 out of the 44 RPPTs. The participation rate of eligible patients in the RPPTs ranged from 48% to 100%, in which 16 RPPTs reported a participation rate higher than 80%, and 14 RPPTs with a participation rate higher than 95%. Of these included participants in the 44 RPPTs, 18%–99% declined randomisation (hence these patients were included in the preference cohort). The randomisation refusal rate was more than 50% in 26 RPPTs.

To assess if a specific patient group accepted randomisation, 35 of the 44 RPPTs reported at least one comparison between randomised and preference cohorts on baseline sociodemographic factors. At least one significant difference between randomised and preference cohorts was found in 20 of the 35 trials. Overall, 38 significant differences were found in 161 sociodemographic comparisons (24%). The proportion of significant findings was not dependent on sample size (smaller trials n<300; 19/85, 22% and larger trials n≥300; 19/76, 25%). Patients with a preference compared with those accepting randomisation were more likely to be older, female, with higher education, employed, Caucasian, not obese, non-smokers, unmarried and experienced with one treatment arm (online supplementary material 3).

Thirty-four of the 44 RPPTs reported at least one comparison between randomised and preference cohorts on clinical baseline characteristics. At least one significant difference was found in 20 of the 34 trials. Overall, 36 significant differences were found in 220 clinical comparisons (16%). The proportion of significant findings was not dependent on sample size (smaller trials n<300; 12/78, 15% and larger trials n≥300; 24/142, 17%). Patients with a preference had more severe clinical problems in seven trials and less severe clinical problems in 10 trials, while in the remaining three trials no consistent pattern could be found (online supplementary material 3).

## Internal validity

The following results concern the influence of patients' preference on internal validity. Information on lost to follow-up in both the randomised and preference cohorts was available in 33 of the 44 RPPTs. For the randomised cohorts, the proportion of individuals lost to follow-up was <10% in 14 trials, 10% to <20% in 9 trials and ≥20% in 10 trials. For the preference cohorts the corresponding numbers of trials were 17, 9 and 7. The mean percentage of participants lost to follow-up was significantly higher in the randomised cohorts (16.1%, SD 16.8%) compared with the preference cohorts (13.3%, SD 14.7%), relative risk (RR 1.3) (95% CI 1.0 to 1.6, p=0.03).

Information on crossovers in both the randomised and preference cohorts was available in 20 of 44 RPPTs. For the randomised cohorts, the proportion of individuals

who crossed over to the other study treatment was <10% in 11 trials, 10% to <20% in 5 trials and ≥20% in 4 trials. For the preference cohorts the corresponding numbers of trials were 14, 5 and 1. The mean percentage of crossovers was significantly higher in the randomised cohorts (14.5%, SD 16.9%) compared with the preference cohorts (6.3%, SD 11.5%), RR 2.6 (95% CI 1.7 to 3.9, p<0.001).

To assess the influence of patients' preference on primary outcomes, for 20 of the 44 RPPTs it was possible to perform reanalyses using standardised effect sizes (figure 1).

Figure 3 shows the magnitude of the experimental treatment effect over the control treatment effect of the randomised and preference cohorts separately using standardised effect sizes. The trials are listed by sample size. A positive experimental treatment effect was seen in 13

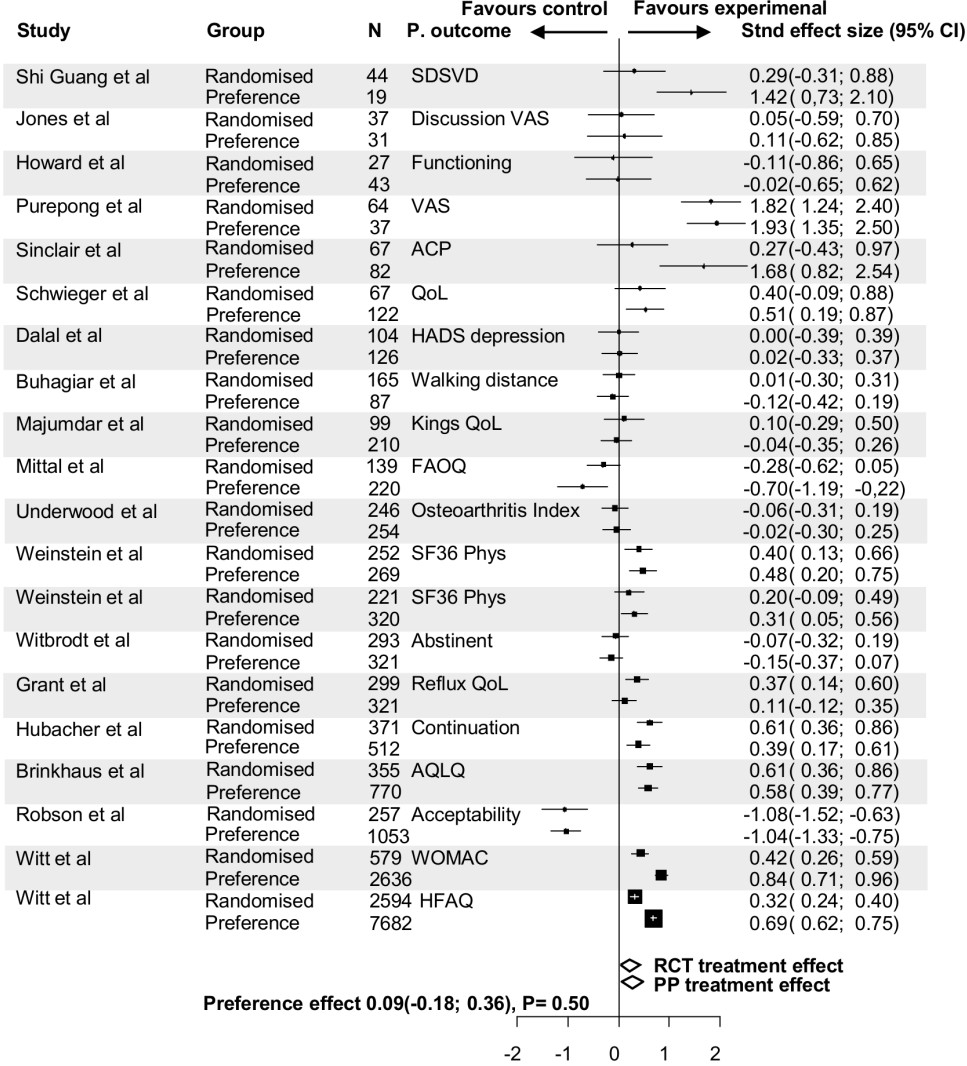

**Figure 3** Forest plot of the preference effect on the primary outcome between the randomised and preference cohorts by comparing the overall treatment effect (standardised effect size) within the randomised cohorts versus the overall treatment effect within the preference cohorts. ACP, advance care planning; AQLQ, Asthma Quality of Life Questionnaire; FAOQ, Foot and Ankle Outcomes Questionnaire; HADS, Hospital Anxiety Depression Scale; HFAQ, Hannover Functional Ability Questionnaire; PP, Patients' preference cohort; QoL, quality of life; RCT, randomised controlled trial; SDSVD, scale of differentiation of syndromes of vascular dementia; SF36, Short Form-36; VAS, visual analogue scale; WOMAC, Western Ontario and McMaster Universities Osteoarthritis Index.

trials. The influence of patients' preference on primary outcomes according to different standardised treatment effects between randomised and preference cohorts was small; in 13 of the 20 trials (65%) this was 0.2 or less (scale −2 to 2), in 5 trials (25%) between 0.21 and 0.5, and in 2 trials (10%) higher than 0.5. Of the 20 RPPTs, the overall mean difference in primary outcome between randomised and preference cohorts was not significantly different, 0.093 (95% CI −0.178 to 0.364, p=0.502) (figure 2). Only two trials showed a significant different treatment effect between the randomised and preference cohorts.[68 69] In both trials the experimental treatment effect was favourable over the control treatment effect in both the randomised and preference cohorts, but the favourable effect of the experimental treatment was significantly greater in the preference cohort. Both RPPTs compared acupuncture versus conservative treatment. In one trial the improvement of the osteoarthritis index in patients with osteoarthritis of the knee or hip was assessed, the other trial assessed the functional ability score in patients with chronic low back pain.

In 7 of these 20 trials, an adjusted primary outcome for baseline confounders was available.[22 32 35 37 60 64 65] In these trials, the mean difference in primary outcome between randomised and preference cohorts was even smaller, −0.026 (95% CI −0.263 to 0.211, p=0.832). In 18 trials (also) a non-adjusted primary outcome was available. Using these outcomes, the mean difference in primary outcomes was 0.228 (95% CI −0.117 to 0.572, p=0.196) (figures 4 and 5).

## DISCUSSION

These study results challenge the current consensus about the hierarchy of study designs. Our results indicate

that patients' preference led to a substantial proportion of patients refusing randomisation (refusal of randomisation was more than 50% in 26 trials), while it did not affect the primary outcome of an RPPT.

Regarding our first hypothesis, it can be concluded that patients' preference does negatively influence participation to RCTs, as demonstrated by the low participation to the randomised cohort in RPPTs. The participation in the RPPTs was remarkably high (ranging from 48% to 100%), improving external validity when compared with the classic RCT (ranging from <0.001% to 40%).[73] Cautiously, it could be argued that a typical patient group characterised by, for example, higher education, Caucasian race and non-obese individuals are more likely to refuse randomisation. In contrast, differences in clinical characteristics showed no consistent pattern in the randomised or preference cohorts. Therefore, not including a patient's preference cohort in a trial could result in a potential loss of inclusions of a specific patient group, further decreasing external validity.

Regarding our second hypothesis, it can be concluded that patients' preference does not significantly affect the primary outcome of an RPPT, as the primary outcomes of patients in the randomised and preference cohorts were similar. Since the aim of an RPPT is to treat patients according to their preference, it can be assumed that the randomised cohort of an RPPT includes patients indifferent to the type of treatment. Subsequently, it is unlikely that outcomes of randomised patients will be biased by treatment preference. Hence, they could be seen as the gold standard. Lost to follow-up and crossovers were significantly higher in the randomised cohort compared with the preference cohort. As a result, the data of the preference cohort could be interpreted more easily than

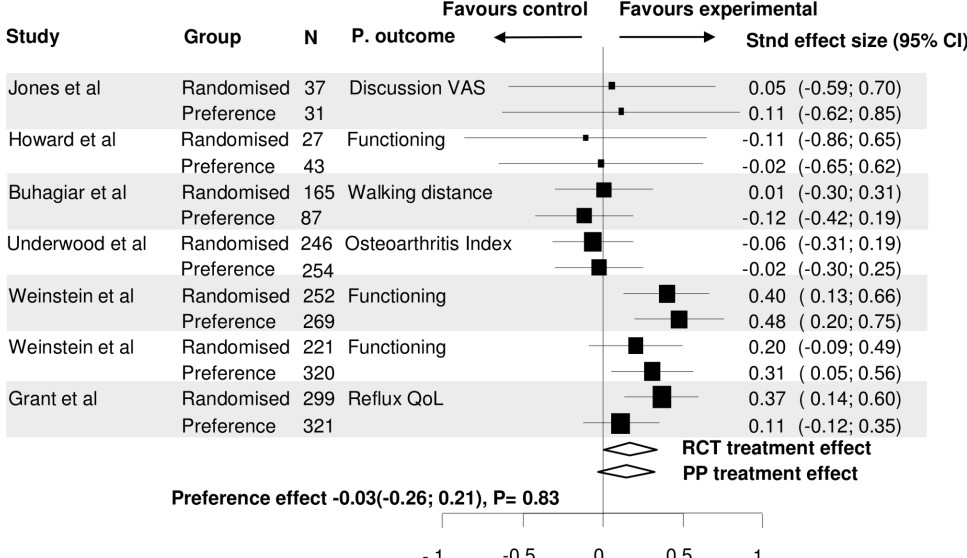

**Figure 4** Forest plot of the preference effect on the primary outcome between the randomised and preference cohorts of trials in which the primary outcome is adjusted for confounders. The overall treatment effect (standardised effect size) within the randomised cohorts was compared with the overall treatment effect within the preference cohorts. QoL, quality of life; RCT, randomised controlled trial; PP, Patients' preference cohort; VAS, visual analogue scale.

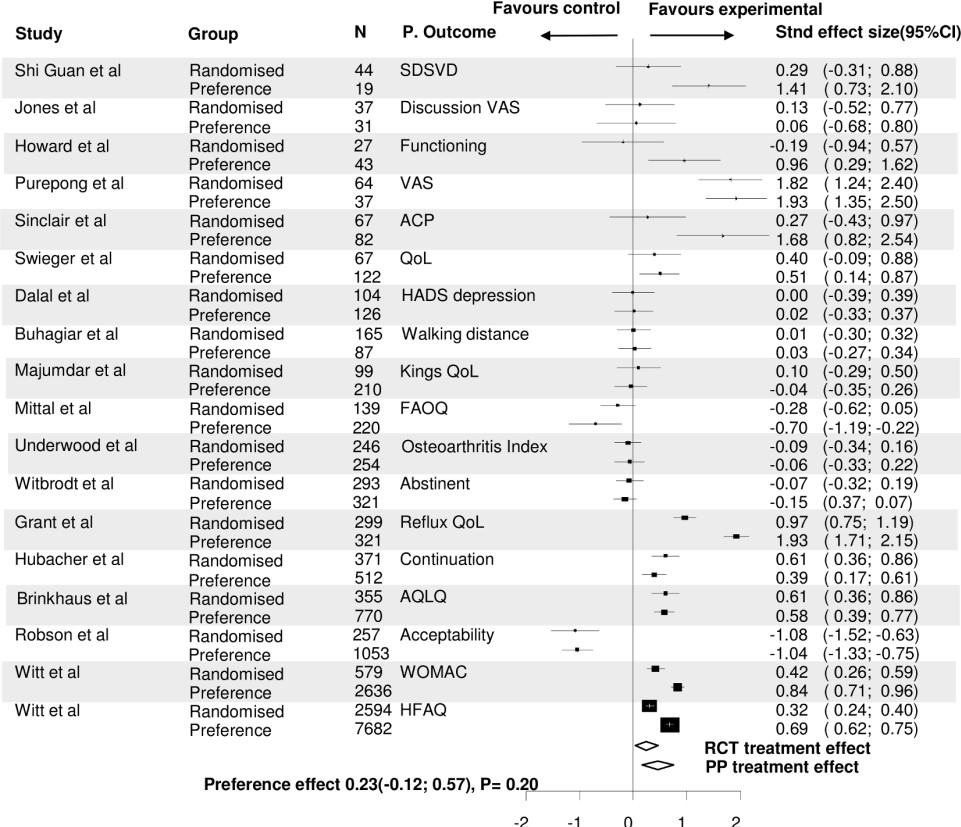

**Figure 5** Forest plot of the preference effect on the primary outcome between the randomised and preference cohorts of trials in which the primary outcome is not adjusted for confounders. The overall treatment effect (standardised effect size) within the randomised cohorts was compared with the overall treatment effect within the preference cohorts. ACP, advance care planning; AQLQ, Asthma Quality of Life Questionnaire; FAOQ, Foot and Ankle Outcomes Questionnaire; HADS, Hospital Anxiety Depression Scale; HFAQ, Hannover Functional Ability Questionnaire; PP, Patients' preference cohort; QoL, quality of life; RCT, randomised controlled trial; SDSVD, scale of differentiation of syndromes of vascular dementia; VAS, visual analogue scale; WOMAC, Western Ontario and McMaster Universities Osteoarthritis Index.

the randomised data. Perhaps, consciously choosing a treatment ensures a certain dedication and tolerance for the treatment.

Our results are strengthened by the previous systematic review of King *et al*, including RPPTs from 1966 to 2004. Based on their results, they also postulated that treatment preference influences the willingness to accept randomisation, and that the evidence of its significant effect on internal validity is low.[4] A possible limitation of their study is that they did not measure patients' preference as specifically as in our analyses, since they also included a minority of two-stage randomised trials, as physician preference.

An RCT is once designed to reliably compare medication to placebo.[74] In the hierarchy of research designs, the results of RCTs are considered to be evidence of the highest grade. Lessons learnt from the history of RCT, and early studies from 1970s and 1980s suggested that observational studies suffer too much from confounders and frequently result in overestimation of treatment effects compared with RCTs.[75 76] Consequently, many experts advocated that results of observational studies should *not* be used for defining evidence-based medical care: '*If the study wasn't randomized, we suggest that you stop reading it and*

*go on to the next article.*'[77] However, two updates of this work including studies between 1985 and 1995 found little evidence that estimates of treatment effects in observational studies are consistently larger than those obtained in RCTs.[78 79] It is suggested that observational studies have methodologically improved over time with the use of a control group, carefully defining inclusion and exclusion criteria, and by better understanding confounders. The fundamental criticism of the RPPT could be that within the preference cohort the unrecognised confounding factors may distort the results. Yet, our results showed that preference cohorts provide valid information comparable with the randomised results.

Today, the classic levels of evidence are subject of debate, as the disadvantages of RCTs have become more insightful in modern practice. In general, patients participating in RCTs are highly selected. Less than 10% of patients participate in trials, partly due to exclusion of patients with a specific treatment preference.[80] This limits the extrapolation of RCT results to patients seen in routine practice. Another consequence is that the majority of trials take several years to be completed. This causes a burden on health research costs, and results in a questionable ethical dilemma. Developments are fast and

the relevance of trials may therefore change over time. Consequently, if an RCT is optimally designed but takes too long, the results will be outdated.

This especially applies when designing a trial in which it can be foreseen that patients' preference will be a prominent factor, for example, in trials comparing treatments of significant different nature (medical vs surgical). Anticipation on the expected patients' preference by eliminating this factor is at the expense of the validity of a lot of RCTs. Especially when patient-centred outcomes are used, one should consider whether the most important patient group has been excluded. Trials must be internally valid, but lack of consideration of external validity causes the widespread underuse of treatments—that showed superiority in RCTs—in routine practice. Moreover, in these situations an RPPT could be the superior design over an RCT.

RPPTs provide unique data on external and internal validity as the patients in the preference cohort are followed according to the same conditions as the patients in the randomised cohorts. A limitation of our review is that interventions and settings between RPPTs were very diverse. On the other hand, because of this diversity, it could also be stated that randomised and preference data often produce similar results in all kinds of settings. Concerning the assessment of external validity, it should be noted that in only a minority of trials the differences in sociodemographic and clinical parameters between the cohorts of an RPPT were evident. Furthermore, in some cases none of the patients in the preference cohort choose the control treatment. In these cases, the treatment effect of the experimental treatment was compared with the control treatment of the randomised cohort. These are not optimal comparisons, but considered to be more appropriate than excluding these data. Moreover, as the idea of RPPTs is a relatively new concept, various terms were used in the inclusion period of this systematic review. In the publication of Walter *et al* in 2017, different concepts were compared and they clearly defined the terms fully randomised patient preference trial and partially randomised patient preference trial. To achieve a 'fully randomised patient preference trial', the preference of all participants should be identified. Therefore, uniform counselling is of crucial importance in RPPTs. The majority of included studies claim to be RPPTs. However, in most of currently included studies, the details of how patients were counselled have not been addressed. As we cannot guarantee that a study identified the preference of all eligible patients, we decided to use the term partially randomised patient preference trials. Another result of the novelty of such a design is that it was not possible to objectively establish the quality of included trials, as there is currently no valid critical appraisal tool to apply for an RPPT. Consequently, our results may have been influenced by the inclusions of flawed trials. In conclusion, RPPTs seem to be a reliable alternative for RCTs, especially in trials comparing treatments of vastly different nature (eg, medical vs surgical)

or using patient-centred outcomes. In case patients' preference can be assumed, RPPT enables faster inclusion of a more representative population improving external validity without compromising internal validity.

**Contributors** KAW and CB designed the study. KAW and PW performed the search. KAW and SvD did the statistical analyses. KAW wrote the first draft with input of CB and WB.

**Funding** The authors have not declared a specific grant for this research from any funding agency in the public, commercial or not-for-profit sectors.

**Competing interests** None declared.

**Patient consent for publication** Not required.

**Provenance and peer review** Not commissioned; externally peer reviewed.

**Data availability statement** Data are available upon reasonable request. All data relevant to the study are included in the article or uploaded as supplementary information.

**ORCID iD**
Karin A Wasmann http://orcid.org/0000-0003-2295-238X

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
