## [Reviewer comments · BMJ Open]

ARTICLE DETAILS

TITLE (PROVISIONAL)	Partially randomised patient preference trials as an alternative design to randomised controlled trials: systematic review and meta-analyses
AUTHORS	Wasmann, Karin; Wijsman, Pieta; van Dieren, Susan; Bemelman, Willem; Buskens, Christianne

VERSION 1 – REVIEW

REVIEWER	Stephen Walter McMaster University, Canada
REVIEW RETURNED	08-May-2019

GENERAL COMMENTS	A key question in interpreting the results of this investigation concerns the detailed conduct of their included studies, specifically about exactly how patients get into the randomised or cohort parts of a given study. Possible study protocols include the following (approximations; details will vary in actual studies): 1) Patients are asked if they have a preference, and if so, they can have it. If not, they are randomised. Preferences are known, de facto, only by the choice of treatments made in the cohort. Cohort members may or may not be aware that the study involves other patients who are being randomised, and that information might also have an impact on preference patterns.2) Patients are invited to be in a randomised trial, but their preferences are not discussed at that stage. If they refuse or decline randomisation, then they get their preferred treatment, but otherwise (i.e. if they are OK with randomisation) they go into the RCT as usual. Preferences (if any) for randomised patients remain unknown.3) As #1, but preferences are also identified for RCT participants. The distinction between #1 and #2 is important. If trials use method #2, we can't assume their patients are refusing randomisation because they actually have a preference. Maybe they don't have an opinion about treatments, and essentially leave the decision up to the physician. Or, perhaps they don't understand the concept of randomisation (many people don't) or the reason for it, and hence distrust leaving their treatment up to some chance mechanism. These possibilities do not imply that they actually have a treatment preference. Even in method #1, we can't assume that patients who become randomised do not have a preference. Maybe they are reluctant
--

to express it in front of intimidating medical professionals who don't take the time to explain the options available...

Finally, in method #3, we would have additional information on preference effects that may affect study outcomes.

Method #3 would constitute a "Fully randomised patient preference design", while #1 and #2 would amount to a "Partially randomised patient preference design". The latter has strong similarities to, but is not totally equivalent to the comprehensive cohort design. See Walter et al., *Statistical Methods in Medical Research* 26, 489, 2017 for some comparisons. I note that several variants of "comprehensive cohort" appear in the authors' Pubmed search, and it would be interesting if they could tell us how many studies were captured by each term: this would elucidate how much overlap (or not) exists between studies of these various types that involve preference information.

But most importantly, I would like to see the authors give us more details of how their studies were done, particularly focussing on the issues I have raised above. How the patients are approached and how they move through the study in this regard could have an important bearing on the study results and their interpretation. In my view, the interpretation of studies in the subgroups above would be very different; can the authors examine their main outcomes in these subgroups? Depending on what is found, this additional information may then require some modification of certain statements in the paper, such as

Patients with a preference for a treatment strategies will be treated accordingly, whereas only those patients without a distinct preference will be randomised in the usual way .

and

it can be expected that many eligible patients decline randomisation due to treatment preference.

I need to be convinced that this is how all the reviewed studies were actually conducted, and that such statements can be validated. (There may be other similar issues in the body of the paper).

Other points:

1) I did not immediately agree this the claim that: "Objective outcomes were defined as a measurement unlikely to be influenced by patients' treatment preference, e.g. mortality". Don't we have be concerned with patient-important outcomes – mortality in particular – and how do we know that such things are unaffected by preferences? There is, for example, the so-called reluctant acquiescence phenomenon of patients in RCT's who get their not-preferred treatment, and this can lead to non-adherence, crossovers, etc, and hence affect outcomes. And you will surely have some of those patients in your included studies.

2) You said: "Of these included participants in the 44 CCTs, 18% to 99% declined randomisation" (page 10) – This is a huge amount of variation, and wonder if you can explain it at all? Is it

	possibly related to how much information was conveyed to potential trial participants, and how it was done? 3) You said: the “mean percentage of participants lost to follow-up was significantly higher in the randomised cohorts” – I found that a bit surprising, because one usually has more contact and control over the patients in an RCT environment. Again, can you explain this finding? 4) What information is available about how the preferred treatment was arrived at in the cohort patients? Can the authors tell us to what extent physicians were involved in helping the patients make the choice, and how they did that? Were decision aids involved? Or were patients simply asked about their preference? Again, these distinctions could have a bearing on how relevant the preference effects are to your study outcomes. General comment: paper is well written and on a very interesting topic. I have no concerns about its other aspects. Minor edits: Page 5, last para: “...participation in RCTs...” Page 8, para 2: “...using a weighted...” Page 8, next para: Wald, not wald.
--	--

REVIEWER	Souraya Sidani Ryerson University, Canada
REVIEW RETURNED	21-May-2019

GENERAL COMMENTS	A. General comments The study addresses a relevant methodological issue related to the performance of alternative trial designs in regard internal and external validity. The argument of the study can be strengthened by explaining how the study extends previous systematic reviews. The methods used are not always clearly described. The results are interesting. The discussion could highlight the mechanisms through which preference influences internal and external validity. The manuscript may need careful editing for clarity. B. Specific comments The title of the paper is not quite consistent with the types of trials included in the review, which seem to be comprehensive cohort trials (as mentioned in the paper). Comprehensive cohort trials is a term not used in the article cited in the Introduction, and may not be quite familiar to many researchers. It may help to first mention the design label as per reference and most widely used, which is partially randomized preference trial, then introduce the comprehensive cohort trial to indicate that these 2 types of trials are the same. Overall, the introduction is rather short and fails to clarify the potential biases to internal and external validity introduced by treatment preferences. Further, the argument for the study can be strengthened by 1) briefly reviewing similar systematic reviews and meta-analyses (e.g. King et al., 2005; Swift and Callahan, 2009; Swift et al., 2013) that synthesized the effects of preferences on attrition and outcomes; and 2) explaining how this study extends previous work – it seems this study is an update of the review done by King et al. (2005), focusing on medical and surgical treatments, as trials conducted in the period from 2005 to 2018 only were included (as the authors explain in a later section).
--

	The contribution of preference to external validity has been reported for RCTs evaluating any type of treatment, including new vs usual ones, and whether or not blinding is applied – this is because the treatments are mentioned or briefly described during the consent process; thus, participants may get an idea about the treatments and may perceive them differently; participants with preferences decline enrollment (and not only participation) to avoid being randomized to non-preferred treatment. The hypothesized mechanism explaining the influence of preference on internal validity could be expanded. It is worth noting that the hypothesized mechanism is applicable to both subjective and objective outcomes, which are affected by adherence to treatment. It is important to clarify up front that “preferences following randomization” (and hence after obtaining consent) is of interest to this study and that such preferences will negatively influence acceptance of randomization and continued participation in a trial – for some methodologists, what the authors refer to as participation represent “early withdrawal”. Please clarify the argument for considering continued participation as an indicator of external validity (rather than internal). What was the reason for including “meta-analyses” in the study and how were their findings accounted for in this study? This raises question about the study design; what is the specific type of review done: systematic review or review of reviews (also called meta-evaluation)? The search terms may have missed some trials and meta-analyses that use other terms for CCT including partially randomized preference trials (e.g. Walter et al., 2017), which may be terms used to classify such trials in some databases. It is a bit hard to follow the logic and the correspondence of the indicators of external validity and the features of the CCTs. It is unclear how “overall participation” rates reflect participation in the preference and random arms of the trial or do the authors mean overall participation in each arm of the CCT? Further, is the comparison on baseline characteristics of participants in the random and preference arms or cohorts indicative of external or internal validity? For some, differences in these cohorts at baseline represent selection bias within treatment groups. Please note that ‘data’ is ‘plural – so use data “were” (instead of “was”). For the indicators of internal validity, it may be worth clarifying early in the respective paragraph that the comparisons between preference and random arms / cohorts were done within the experimental and control groups – of course, if this is correct. Otherwise, please revise the statements for clarity. How appropriate is it to compare the experimental treatment in the preference arm to the control treatment in the random arm, when no participant was allocated by preference to the control treatment? Please justify this decision. Evidence suggests that participants allocated to control or comparison treatment by preference exhibit more improvement in outcomes than those randomized to the control treatment. It is a bit hard to understand the plan for data analysis – what is the focus of the analysis: within or between group comparisons and what groups were included in what analysis? The statement about patient and public involvement is unclear.
--	--

	The presentation of the results is much easier (than the rest of the paper) to follow, except is not always clear what effect size is reported on: within or between group? The authors may want to compare their findings to other systematic reviews that showed a small but significant effect of preference on the outcomes, and to explore the wider (i.e., outside medicine) methodological literature that proposed how preference could influence internal and external validity, and integrate relevant mechanisms responsible for the influence of preferences in the discussion. The discussion related to the use of observational study may not be quite relevant to the concept of preference – observational studies usually do not account for preferences.
--	--

VERSION 1 – AUTHOR RESPONSE

Reviewer: 1

Reviewer Name: Stephen Walter

Institution and Country: McMaster University, Canada Please state any competing interests or state 'None declared': None declared.

A key question in interpreting the results of this investigation concerns the detailed conduct of their included studies, specifically about exactly how patients get into the randomised or cohort parts of a given study. Possible study protocols include the following (approximations; details will vary in actual studies):

1. Patients are asked if they have a preference, and if so, they can have it. If not, they are randomised. Preferences are known, de facto, only by the choice of treatments made in the cohort. Cohort members may or may not be aware that the study involves other patients who are being randomised, and that information might also have an impact on preference patterns.
2. Patients are invited to be in a randomised trial, but their preferences are not discussed at that stage. If they refuse or decline randomisation, then they get their preferred treatment, but otherwise (i.e. if they are OK with randomisation) they go into the RCT as usual. Preferences (if any) for randomised patients remain unknown.
3. As #1, but preferences are also identified for RCT participants.

The distinction between #1 and #2 is important. If trials use method #2, we can't assume their patients are refusing randomisation because they actually have a preference. Maybe they don't have an opinion about treatments, and essentially leave the decision up to the physician. Or, perhaps they don't understand the concept of randomisation (many people don't) or the reason for it, and hence distrust leaving their treatment up to some chance mechanism. These possibilities do not imply that they actually have a treatment preference.

Answer 1:

We completely agree with this comment. Unfortunately, most of the included studies do not include extensive details of how patients were counselled, nor do they provide details of how (and when) patients were included in the randomised cohort or in the patient preference cohort. As we aimed to include all papers that intended to treat patients with a preference accordingly, and only randomised those patients without a distinct preference, we have excluded two-stage randomised designs. In those trials participants are initially randomised into two groups. In the first group they are offered a

choice of treatment, while in the second group they are randomized to treatment.[1] Or the alternative of Rücker, in which the participants randomised to the preference group who do not have a strong preference are in a second stage randomised to a treatment.[2] Still the difference between method #1 and 2 are of major importance and when not completely identified it will lead to bias of interpreting results, as suggested by the reviewer. We have added this in the discussion (page 16) and addressed it as a limitation of the study in strength and limitation (page 4).

“Uniform counselling is of crucial importance in RPPTs, which has not been standardly reported in the included studies.”

“Moreover, as the idea of RPPTs is a relatively new concept, various terms were used in the inclusion period of this systematic review. In the publication of Walter et al in 2017 different concepts were compared and they clearly defined the terms fully randomised patient preference trial and partially randomised patient preference trial. To achieve a ‘fully randomised patient preference trial’ preference of all participants should be identified. Therefore, uniform counselling is of crucial importance in RPPTs. The majority of included studies claim to be randomised patient preference trials. However, in most of currently included studies, the details of how patients were counselled has not been addressed. As we can’t guarantee that a study identified the preference of all eligible patients, we decided to use the term partially randomised preference trials.”

Even in method #1, we can’t assume that patients who become randomised do not have a preference. Maybe they are reluctant to express it in front of intimidating medical professionals who don’t take the time to explain the options available...

Answer 2:

Indeed. Thank you for pointing out these subtle differences. To make this more clear for the readers of the manuscript, we have changed the term comprehensive cohort trial (CCT) into partially randomised patient preference trial (RPPT), as we can’t just assure all studies included were fully randomised patient preference trials/CCTs. This issue has now been also addressed in the discussion (page 16, see answer 1).

Finally, in method #3, we would have additional information on preference effects that may affect study outcomes.

Method #3 would constitute a “Fully randomised patient preference design”, while #1 and #2 would amount to a “Partially randomised patient preference design”. The latter has strong similarities to, but is not totally equivalent to the comprehensive cohort design. See Walter et al., *Statistical Methods in Medical Research* 26, 489, 2017 for some comparisons. I note that several variants of “comprehensive cohort” appear in the authors’ Pubmed search, and it would be interesting if they could tell us how many studies were captured by each term: this would elucidate how much overlap (or not) exists between studies of these various types that involve preference information.

Answer 3:

Fifty-eight studies could be included, of which just 8 were captured with the term comprehensive cohort. Most studies were found under the term randomised patient preference trial (n = 17), followed by patient preference trial (n = 15), fully randomised patient preference trials (n = 5), partially randomised preference trial (n = 1), Zelen randomised controlled trial (n = 1). Additionally, 11 studies were found by other terms as preference tolerant randomised controlled trial or pragmatic randomised controlled trial. We believe this illustrates that the terms fully and partially randomised patient preference trials were not as well established in the study search period. We have added this to the discussion (page 15).

But most importantly, I would like to see the authors give us more details of how their studies were done, particularly focussing on the issues I have raised above. How the patients are approached and how they move through the study in this regard could have an important bearing on the study results and their interpretation. In my view, the interpretation of studies in the subgroups above would be very different; can the authors examine their main outcomes in these subgroups? Depending on what is found, this additional information may then require some modification of certain statements in the paper, such as

Patients with a preference for a treatment strategies will be treated accordingly, whereas only those patients without a distinct preference will be randomised in the usual way .

and

it can be expected that many eligible patients decline randomisation due to treatment preference.

I need to be convinced that this is how all the reviewed studies were actually conducted, and that such statements can be validated. (There may be other similar issues in the body of the paper).

Answer 4:

We have only included those studies that were described in their own paper as a comprehensive cohort design, or (fully/partially) randomised patient preference trial. Furthermore, it was the reason to exclude the stage-2 trials, as within these design it is not aimed to treat patients according to their preference. However, due to lack of details we have been unable to objectify how patients were exactly counselled or to measure the strength of preference of patients. We have selected the papers to the best of our abilities. For this study we considered the three methods as described above. Nonetheless, we could not firmly discriminate between the three methods based on the details provided in the papers. However, we believe the aim of this paper is to bring the alternative design to the attention of the broader readership of the BMJ open, of whom the majority will not have a background in epidemiology. Therefore, we think that a subdivision (if possible) would make interpretation unnecessary difficult.

Finally, we have nuanced the statements, as suggested by the reviewer. (page 5, 13):

“Several alternative names and approaches have been proposed as alternative designs to preclude the influence of patients’ preference on validity: a partially randomised preference trial (RPPT), a comprehensive cohort trial, a patient preference trial, and more. In general the aim of these designs is to treat patients with a preference for a treatment strategies accordingly, whereas only those patients without a distinct preference will be randomised in the usual way.”

Other points:

1) I did not immediately agree this the claim that: “Objective outcomes were defined as a measurement unlikely to be influenced by patients’ treatment preference, e.g. mortality”. Don’t we have be concerned with patient-important outcomes – mortality in particular – and how do we know that such things are unaffected by preferences?

There is, for example, the so-called reluctant acquiescence phenomenon of patients in RCT’s who get their not-preferred treatment, and this can lead to non-adherence, crossovers, etc, and hence affect outcomes. And you will surely have some of those patients in your included studies.

Answer 5:

We totally agree. Originally, it was the idea to do a sensitivity analysis between studies with subjective and objective outcomes parameters. It soon became clear that this dichotomy was subjectively assessed. We agree that patient-important outcomes are subjective and will be influences by

preference. Indeed, mortality can be affected via the principle of reluctant acquiescence phenomenon. In the end it is impossible to determine objective parameters, and we have decided to remove it.

2) You said: “Of these included participants in the 44 CCTs, 18% to 99% declined randomisation” (page 10) – This is a huge amount of variation, and wonder if you can explain it at all? Is it possibly related to how much information was conveyed to potential trial participants, and how it was done?

Answer 6:

You are right. Unfortunately, detailed method of counselling has not been described in the majority of studies. Therefore, we can't correlate results to the amount of information provided to the potential participant of various studies. This has been described as a limitation as stated above (page 4 and 16). We hypothesize that the percentage of patients declining randomisation can be influenced by counselling, but it might also be associated to studies comparing treatments of significant different nature. For example, more patients might decline randomisation when the study analyses a medical to a surgical treatment when compared to the randomisation in a non-superiority trial between two established medical treatments.

3) You said: the “mean percentage of participants lost to follow-up was significantly higher in the randomised cohorts” – I found that a bit surprising, because one usually has more contact and control over the patients in an RCT environment. Again, can you explain this finding?

Answer 7:

We were surprised as well, and we can only speculate on the cause of this finding. Both preference and RCT cohorts were treated according to the same follow-up protocol. We hypothesise that acknowledging patients' preferences might increase the adherence to a treatment by patients. This is confirmed in the meta-analyses of Swift and Callahan et al[3].

4) What information is available about how the preferred treatment was arrived at in the cohort patients? Can the authors tell us to what extent physicians were involved in helping the patients make the choice, and how they did that? Were decision aids involved? Or were patients simply asked about their preference? Again, these distinctions could have a bearing on how relevant the preference effects are to your study outcomes.

Answer 8:

We fully agree. Unfortunately the information provided in the majority of papers does not allow us to provide this valuable specific information. This has now been addressed in the limitations as stated in answer 1 (page 4 and 16).

General comment: paper is well written and on a very interesting topic. I have no concerns about its other aspects.

Minor edits:

Page 5, last para: “...participation in RCTs...”

Page 8, para 2: “...using a weighted...”

Page 8, next para: Wald, not wald.

Answer:

We apologize for the typos and we have adjusted them accordingly.

Overall, we would like the reviewer for his insightful comments. The revisions have definitely improved the manuscript.

Reviewer: 2

Reviewer Name: Souraya Sidani

Institution and Country: Ryerson University, Canada Please state any competing interests or state 'None declared': None declared

A. General comments

- The study addresses a relevant methodological issue related to the performance of alternative trial designs in regard internal and external validity.
- The argument of the study can be strengthened by explaining how the study extends previous systematic reviews.
- The methods used are not always clearly described.
- The results are interesting.
- The discussion could highlight the mechanisms through which preference influences internal and external validity.
- The manuscript may need careful editing for clarity.

B. Specific comments

Title

The title of the paper is not quite consistent with the types of trials included in the review, which seem to be comprehensive cohort trials (as mentioned in the paper).

Answer 1:

As described in comments 3 of reviewer 1, the term comprehensive cohort study design could be confusing, and has not been consistently used throughout the included studies. We have therefore decided to use the term partially randomised patient preference trial, and we have adjusted the title.

Introduction

Comprehensive cohort trials is a term not used in the article cited in the Introduction, and may not be quite familiar to many researchers. It may help to first mention the design label as per reference and most widely used, which is partially randomized preference trial, then introduce the comprehensive cohort trial to indicate that these 2 types of trials are the same.

Answer 2:

The introduction has been rewritten to make this more clear (page 5): "Throughout the years, several approaches, using various names, have been proposed as alternative designs to preclude the influence of patients' preference on validity: a partially randomised preference trial (RPPT), a comprehensive cohort trial, a patient preference trial, and more.(included ref Walters here) In general the aim of these designs is to treat patients with a preference for a treatment strategies accordingly, whereas only those patients without a distinct preference will be randomised in the usual way."

Overall, the introduction is rather short and fails to clarify the potential biases to internal and external validity introduced by treatment preferences.

Answer 3:

Thank you very much for your suggestions. We have described the potential biases introduced by treatment preferences more extensively (page 5):

"Randomised controlled trials (RCTs) are suggested to provide the most reliable evidence for treatment efficacy. However, participants are no passive recipients of interventions. Patients with a treatment preference may decline enrolment to avoid being randomised to their non-preferred

treatment. Consequently, treatment preferences can decrease the generalizability of RCTs results to the clinical population (i.e. reduce external validity). Additionally, trials comparing experimental vs standard treatment, are likely to include patients preferring experimental treatment, as trial participation is not needed for patients preferring standard treatment, further reducing external validity. Internal validity may be reduced as randomisation to the (non-) preferred strategy could influence adherence to treatment protocol and study outcomes. Subjective study outcomes can directly be affected by treatment preference, whereas objective outcomes most likely may be affected indirectly via adherence (so called reluctant acquiescence phenomenon). Especially for an unblinded trial comparing treatments of significant different nature (e.g. medical vs surgical) the RCT could be an inappropriate design.”

Further, the argument for the study can be strengthened by 1) briefly reviewing similar systematic reviews and meta-analyses (e.g. King et al., 2005; Swift and Callahan, 2009; Swift et al., 2013) that synthesized the effects of preferences on attrition and outcomes; and 2) explaining how this study extends previous work – it seems this study is an update of the review done by King et al. (2005), focusing on medical and surgical treatments, as trials conducted in the period from 2005 to 2018 only were included (as the authors explain in a later section).

Answer 4:

Thank you very much, we have added the references, and have more extensively addressed the strength that this paper is useful for current daily clinical practice (page 5,6):

“The two previous systematic reviews addressing influence of preference on validity, concluded that this influence was limited. (King, and included ref Swift and Callahan here). However, one review only included studies addressing psychotherapy, and the other dates from 2005. So far, the value of the RPPT remains unclear, nor has it been addressed in the Oxford Levels of Evidence (CEBM). The aim of the study was to assess the influence of patients’ preference following randomisation in current daily clinical practice, by comparing randomised cohorts with preference cohorts within all RPPTs published since 2005.”

The contribution of preference to external validity has been reported for RCTs evaluating any type of treatment, including new vs usual ones, and whether or not blinding is applied – this is because the treatments are mentioned or briefly described during the consent process; thus, participants may get an idea about the treatments and may perceive them differently; participants with preferences decline enrollment (and not only participation) to avoid being randomized to non-preferred treatment.

Answer 5:

We totally agree and thank you for highlighting this. A potential reduction in external validity even applies to blinded trials. We have adjusted the introduction (page 5), as stated above in answer 3.

The hypothesized mechanism explaining the influence of preference on internal validity could be expanded. It is worth noting that the hypothesized mechanism is applicable to both subjective and objective outcomes, which are affected by adherence to treatment.

Answer 6:

Thank you, we have added this in the introduction (page 5), as stated above in answer 3.

It is important to clarify up front that “preferences following randomization” (and hence after obtaining consent) is of interest to this study and that such preferences will negatively influence acceptance of randomization and continued participation in a trial – for some methodologists, what the authors refer to as participation represent “early withdrawal”. Please clarify the argument for considering continued participation as an indicator of external validity (rather than internal).

We agree, and definitely think early withdrawal could be affected by patient preferences. We hypothesise that acknowledging patients' preferences might increase the adherence to a treatment by patients (thus internal validity). To clarify this we have adjusted the second hypothesis in the introduction (page 6):

"Patients' preferences will influence adherence and outcomes in RCTs, decreasing internal validity."

Furthermore, as addressed in the methods section, the main outcomes are the difference in external validity (participation and baseline characteristics) and internal validity (lost to follow-up, cross-over and the primary outcome) between the random cohort versus the preference cohort within each RPPT (page 8).

Methods

What was the reason for including "meta-analyses" in the study and how were their findings accounted for in this study? This raises question about the study design; what is the specific type of review done: systematic review or review of reviews (also called meta-evaluation)?

Answer 7:

We apologize for the confusion. Meta-analyses were not included in the search strategy. We just wanted to start our methods by stating the study design. Overall we have made more subheadings to make the methods section easier to read . We have adjusted it in the methods:

"Design

A systematic review and meta-analyses of RPPTs was conducted."(page 7)

The search terms may have missed some trials and meta-analyses that use other terms for CCT including partially randomized preference trials (e.g. Walter et al., 2017), which may be terms used to classify such trials in some databases.

Answer 8:

By the use of the term 'preference' and 'randomised' in our search we believe our search was accurate. Of the fifty-eight studies included, just 8 were captured with the term comprehensive cohort. Most studies were found under the term randomised patient preference trial (n = 17), followed by patient preference trial (n = 15), fully randomised patient preference trials (n = 5), partially randomised preference trial (n = 1), Zelen randomised controlled trial (n = 1). Additionally, 11 studies were found by other term as preference tolerant randomised controlled trial or pragmatic randomised controlled trial.

It is a bit hard to follow the logic and the correspondence of the indicators of external validity and the features of the CCTs. It is unclear how "overall participation" rates reflect participation in the preference and random arms of the trial or do the authors mean overall participation in each arm of the CCT?

Answer 9:

We adjusted it in the methods to (page 8):

"To analyse whether patients' preference influenced external validity, data were extracted on participation rates in the randomised and preference cohort."

Further, is the comparison on baseline characteristics of participants in the random and preference arms or cohorts indicative of external or internal validity? For some, differences in these cohorts at baseline represent selection bias within treatment groups.

Answer 10:

We agree that it also indicates selection bias. However, the aim of the analyses was to check if a specific characterised patient group accepted randomisation. In that context we defined it as external validity. Following the analyses, cautiously it could be argued that a typical patient group characterised by e.g. higher education, Caucasian race, and non-obese individuals are more likely to refuse randomisation. In contrast, differences in clinical characteristics (potential selection bias) showed no consistent pattern in the randomised or preference cohorts. Therefore, not including a patients' preference cohort in a trial could result in a potential loss of inclusions of a specific patient group, further decreasing external validity (page 12).

Please note that 'data' is 'plural – so use data “were” (instead of “was”).

Answer 11:

Thank you very much for pointing this out, we have adjusted it in the manuscript.

For the indicators of internal validity, it may be worth clarifying early in the respective paragraph that the comparisons between preference and random arms / cohorts were done within the experimental and control groups – of course, if this is correct. Otherwise, please revise the statements for clarity.

Answer 12:

We have adjusted the sentences to make clearer that the comparisons are done within the RPPTs (page 9):

“Following, these outcomes were compared between the randomised and preference cohorts within RPPTs.”

“To compare the primary outcomes between the randomised and preference cohorts within RPPTs, the outcome effects were compared between the randomised cohort and the preference cohort.”

How appropriate is it to compare the experimental treatment in the preference arm to the control treatment in the random arm, when no participant was allocated by preference to the control treatment? Please justify this decision. Evidence suggests that participants allocated to control or comparison treatment by preference exhibit more improvement in outcomes than those randomized to the control treatment.

Answer 13:

We believe more improvement can be expected in the experimental treatment of the preference arm. Especially when nobody choose control treatment, suggesting participants are real 'believers'. However, after extensive consultation with our epidemiologist SvD, the only other alternative to exclude these trials seemed even less appropriate. We have addressed it as a limitation in the discussion (page 16):

“Furthermore, in some cases none of the patients in the preference cohort choose the control treatment. In these cases, the treatment effect of the experimental treatment was compared with the control treatment of the randomised cohort. These are not optimal comparisons, but considered to be more appropriate then excluding these data.”

It is a bit hard to understand the plan for data analysis – what is the focus of the analysis: within or between group comparisons and what groups were included in what analysis?

Answer 14:

The main focus of the paper is on the between group treatment effects (patient-preference group and randomised group). We have now clarified this in the data analysis section of the paper.

The statement about patient and public involvement is unclear.

Answer 15:

As patients or the public were not directly involved, we have adjusted it to:

“There was no direct involvement of patients or the public in the development of the research question, selection of the outcomes measures, design and implementation of the study, or interpretation of the results.” (page 10)

Results

The presentation of the results is much easier (than the rest of the paper) to follow, except is not always clear what effect size is reported on: within or between group?

Answer 16:

The paper is about between group effects and addresses treatment effects of the preference cohort versus the randomised cohort, which is clarified throughout the paper.

Discussion

The authors may want to compare their findings to other systematic reviews that showed a small but significant effect of preference on the outcomes, and to explore the wider (i.e., outside medicine) methodological literature that proposed how preference could influence internal and external validity, and integrate relevant mechanisms responsible for the influence of preferences in the discussion.

Answer 17:

The aim of this study is to present this rather new concept to the general medical clinician in the field. We have addressed the meta-analyses of Swift and Callahan in the introduction (page 5). However, the meta-analyses of Swift and Callahan focused on psychotherapy. As this is a update of the study of King et al, we feel that it is more relevant to discuss the paper of King et al.

We agree that it is very interesting to read the wider methodological literature. We have learned a lot, but we feel that including reviews outside clinical medicine would be out of the scope of this paper.

The discussion related to the use of observational study may not be quite relevant to the concept of preference – observational studies usually do not account for preferences.

Answer 18:

Supplementary to the previous answer, we think it helps readers without an epidemiologic background to read about the rise of RCTs and the developments in observational studies. The take home message of the paper is that one should choose the most suitable design for the research question (and patients) instead of just believing that an RCT is always the holy grail.

References:

1 Wennberg JE, Barry MJ, Fowler FJ, et al. Outcomes research, PORTs, and health care reform. *Ann N Y Acad Sci* 1993;703:52–62. doi:10.1111/j.1749-6632.1993.tb26335.x

2 Rücker G. A two-stage trial design for testing treatment, self-selection and treatment preference effects. *Stat Med* 1989;8:477–85. <http://www.ncbi.nlm.nih.gov/pubmed/2727471> (accessed 3 Jul 2019).

3 Swift JK, Callahan JL. The impact of client treatment preferences on outcome: a meta-analysis. *J Clin Psychol* 2009;65:368–81. doi:10.1002/jclp.20553

VERSION 2 – REVIEW

REVIEWER	Souraya Sidani Professor and Canada Research Chair Ryerson University Canada
REVIEW RETURNED	15-Jul-2019

GENERAL COMMENTS	In the revised manuscript, the authors addressed previous reviewers' comments. They clarify points, clearly described methods, and identified limitations of available data (in particular the differences in preference trial terminology and design, and the limited information on when and how exactly were preferences assessed. It may help to edit the revised manuscript to rectify some grammatical mistakes (e.g. "it can be conclude" - should be "concluded"). Overall, this is a very important contribution to the methodological literature.
--